# Experimental Investigation and Modeling of Film Flow Corrosion

**Marius Ciprian Ilie** [1,2], **Ioana Maior** [3], **Cristian Eugen Raducanu** [1], **Iuliana Mihaela Deleanu** [1,*], **Tanase Dobre** [1,4,*] **and Oana Cristina Parvulescu** [1]

1. Chemical and Biochemical Engineering Department, University Politehnica of Bucharest, 1-7 Gheorghe Polizu St., 011061 Bucharest, Romania; ciprian.ilie@totalenergies.com (M.C.I.); cristianrdcn1@yahoo.com (C.E.R.); oana.parvulescu@yahoo.com (O.C.P.)
2. TotalEnergies Marketing Romania SA, 4 Vasile Alecsandri St., 010639 Bucharest, Romania
3. Inorganic Chemistry, Physical Chemistry and Electrochemistry Department, University Politehnica of Bucharest, 1-7 Gheorghe Polizu St., 011061 Bucharest, Romania; ioana.maior@upb.ro
4. Technical Sciences Academy of Romania, 26 Dacia Boulevard, 030167 Bucharest, Romania
* Correspondence: iuliana.deleanu@upb.ro (I.M.D.); tghdobre@gmail.com (T.D.)

**Abstract:** The paper focuses on the experimental investigation and mathematical modeling of the corrosion of steel when a film of water flows over its surface. The experimental monitoring of corrosion dynamics in the flowing film was carried out using a laboratory pilot model, exploited in such a way as to obtain data necessary to identify some characteristic parameters of the mathematical model of this problem. The mathematical model of the case takes into account the transfer of oxygen through the liquid film flowing on the surface of the corroding plate where the chemical surface processes characteristic of corrosion occur (dissolution of $Fe$, oxidation of $Fe^{2+}$ to $Fe^{3+}$, formation of surface deposit, etc.). Experimental measurements were used to identify the parameters of the mathematical model, especially the reaction constant of the $Fe$ dissolution rate and the surface oxidation yield of $Fe^{2+}$ to $Fe^{3+}$. Calculation of the correlation coefficients for the apparent constant surface reaction rate and process factors showed that they correlate strongly and non-linearly with the *Reynolds number* (*Re*) of the film flow, with the cumulative flow duration, and with the cumulative standby time of the experiments. Using the dynamics of the resistance to the transfer of oxygen through the rust film and the dynamics of its thickness resulting from the specific flow of rust deposition, the apparent oxygen diffusion coefficient through the rust film formed on the plate was expressed.

**Keywords:** oxygen transfer; film flow; corrosion kinetics; mathematical modeling; steel corrosion; process dynamics





## 1. Introduction

Corrosion is a natural phenomenon in which the flow, heat, and mass transfer of oxygen and water from the surrounding environment to a solid structure (usually a metal) causes it to chemically deteriorate due to a surface reaction with oxygen and the presence of water. We know that uncontrolled corrosion has led to many disasters [1–3], being unfairly considered as a destructive natural phenomenon, similar to those of unpredictable natural origin. Over the past 20 years, the world has experienced approximately 500 major weather-related disasters, with total normalized losses of more than USD 170 billion annually. And, among all these disasters, a significant number of events, even of a small size, have the uncontrolled evolution of the corrosion of some metallic structures as an explanation. In other words, corrosion is a controllable process that has a price that must be carefully considered by all developers of industrial and civil metal structures, anywhere in the world. In this context, we note that, in the design of industrial metallic systems, there are generalized rules regarding corrosion control [4,5]. For example, for chemical plants, corrosion control costs are around 10–12% of the investment value, plus annual corrosion

control maintenance in the range of 1–2% of the present investment value. In the control of metallic corrosion, experimental investigation and modeling have an extremely important role, because they are strongly oriented towards the identification and analysis of the elementary processes that determine the formation and installation of corrosion. As for the characteristics of elementary corrosion processes, they are determined by the presence of oxygen and water at the contact surface of the metal with the environment, with a strong specificity, and influenced by the exact same environment. In the case of atmospheric corrosion, despite multiple factors, many of them with random action, the fundamental ones are film corrosion [6] and droplet corrosion [7–9].

*Film corrosion* occurs during rain, when a water film is formed on the metal surface in a downward flow. It controls the oxygen transfer from the air to the solid surface, where the chemical (electrochemical) dissolution of the metal takes place. It is a complex process of diffusion, mass transfer, and chemical reaction in which oxygen is the diffusible species, whereas the reactive species are, at the metallic surface, water, oxygen, and metal. *Droplet corrosion* is a sum of three extremely complex phenomena that includes: proper condensation as a simultaneous process of heat and mass transfer when water vapor from the environment condenses into droplets on the metal surface as the first phenomenon; the transfer and diffusion of oxygen through the drop gives a corrosion reaction on the metal surface as the second phenomenon; and the droplets evaporate and the surface dries, which is in fact a simultaneous process of heat and mass transfer and is the third phenomena. Figure 1 shows that, regardless of the method of steel protection, the absence of film and control of droplet corrosion leads to irreversible degradation of equipment exposed to the atmosphere (tanks from an oil terminal in the present case).

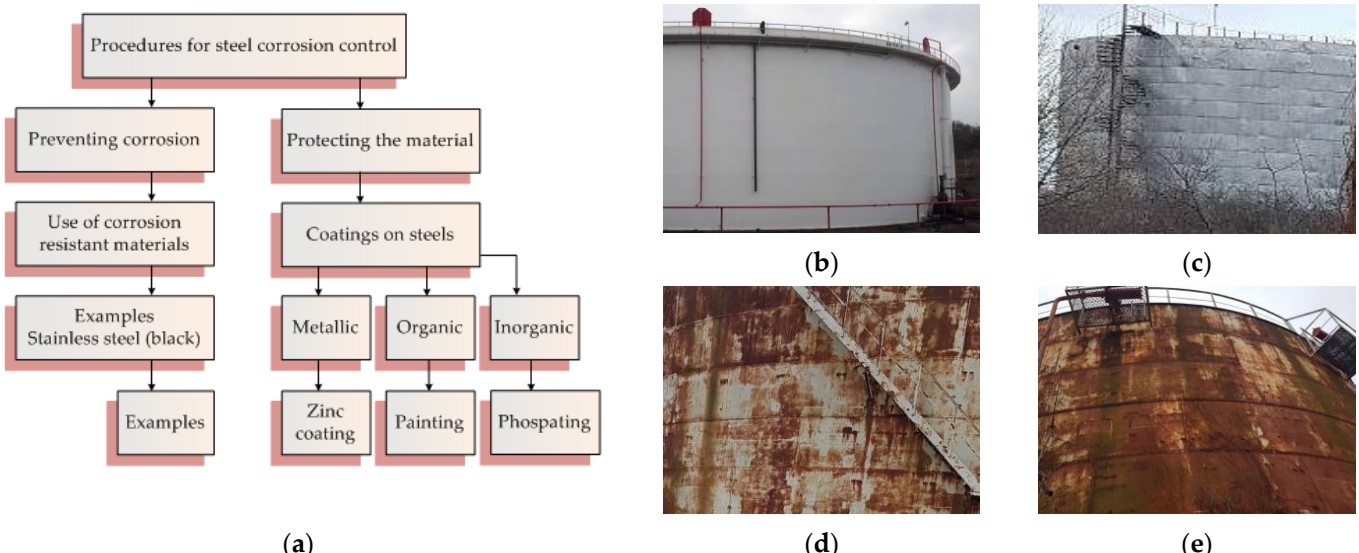

**Figure 1.** Procedure for steel corrosion control and results of uncontrolled of film and drops corrosion of some tanks from an oil terminal: (**a**) stages of corrosion control; (**b**,**c**) tanks at the beginning of usage; (**d**,**e**) tanks after years of usage.

Developed in the form of film or droplets, atmospheric corrosion is responsible for the massive degradation of metallic structures, and knowledge of their elementary process dynamics [10–12] would contribute to achieving two major objectives: in-depth knowledge of the phenomenon, so that it can be investigated through modeling and simulation, and the choice of the best anti-corrosion protection method (Figure 1).

In Figure 2 where the momentary elementary processes of corrosion by water droplets are presented, we observe:

1.　An electrochemical process that identifies an anodic and a cathodic location, between which, the transfer of electrons takes place due to the corresponding cathodic and anodic reactions;
2.　A mass transfer process through which oxygen reaches the liquid phase ($O_{2g} \rightarrow O_2$), to maintain the cathodic and anodic process;
3.　A chemical reaction process, partly dependent on $O_2$, which leads to the formation of rust on the steel surface ($Fe^{2+} + 2OH^- \rightarrow Fe(OH)_2$, $2Fe(OH)_2 + \frac{1}{2}O_2 + H_2O \rightarrow 2Fe(OH)_3$).

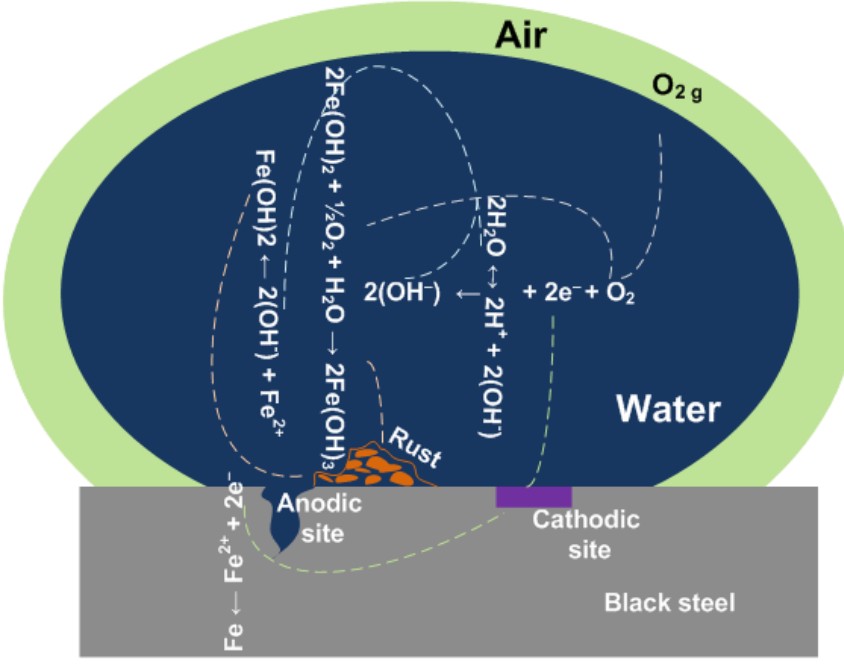

**Figure 2.** Physical and chemical processes at atmospheric steel corrosion in a water drop.

If the ionization of water is expressed as a hydrogen ($H^+$) ion generator as pointed out in Equation (1), it uses the anodic reaction (metal dissolution) in the form of Equation (2) and the cathodic reaction in the form of Equation (3); then, by summing them, Equation (4) is obtained.

$$4H_2O_{l)} \rightarrow 4H^+{}_{(l)} + 4OH^-{}_{(l)} \tag{1}$$

$$2Fe_{(s)} \rightarrow 2Fe^{2+}{}_{(l)} + 4e^- \tag{2}$$

$$O_{2(l)} + 4H^+{}_{(l)} + 4e^- \rightarrow 2H_2O_{(l)} \tag{3}$$

$$2Fe_{(s)} + 2H_2O_{(l)} + O_{2(l)} \rightarrow 2Fe^{2+}{}_{(l)} + 4OH^-{}_{(l)} \tag{4}$$

From the point of view of chemical engineering, Equation (4) expresses a solid–liquid reaction, where it is obvious that the reaction itself (the sum of Equations (2) and (3)) takes place in active centers, as it does in heterogeneous catalysis. In this case, each reaction center consists of anodic and cathodic components. A consequence of this consideration is that this case can be treated kinetically with the methods specific to heterogeneous solid–liquid reactions.

In Equation (4), the kinetics of the process depends on the oxygen concentration in the liquid. This is strongly influenced by the transfer of oxygen from gas to liquid and from there to the cathodic side of the corrosion site. Now, it must be added that the mentioned heterogeneous process is, after oxygen, also under the influence of the local formation of

rust points (centers). This could be associated with each corrosion center as shown by the heterogeneous reaction shown in Equation (5). It follows from Equation (4), to which the oxidation of $Fe^{2+}$ to $Fe^{3+}$ and the precipitation of the metal oxide are added, what are processes that occur outside or in the vicinity of the corrosion site.

$$Fe_{(s)} + \frac{3}{2}O_{2\,(l)} + (z+3)H_2O \rightarrow 2Fe(OH)_3 zH_2O \rightarrow Fe_2O_3 zH_2O + 3H_2O \qquad (5)$$

When water is on the corrosion surface as a film flow (a case of rain), in addition to the processes mentioned, an erosional entrainment of metallic oxides ($Me_2O_3 \cdot zH_2O$) agglomerates forming on the corrosion surface can occur. In droplet corrosion, there is no training but there is an increase (mass, volumetric (spatial)) in $Me_2O_3 \cdot zH_2O$ accumulation. For this case, we can add, as a very interesting phenomenon, an observable drying of the environment adjacent to the $Me_2O_3 \cdot zH_2O$ rust due to the evaporation of the droplets and the local heating, caused by heat transfer around the corrosion center. Looking at Equation (5) from the point of view of the heterogeneous kinetics process, we conclude that it can be expressed by the transfer rate of at least one species, $Fe$ or $O_2$, in the considered case. It is thus obvious that we must express the consumption rate of metal ($Me$) due to the film or droplet corrosion. The present paper will focus on these two issues. It begins with exposing the case of corrosion of steel when we have a water flowing film on its surface. In approaching this problem, the path followed was that of experimental investigation, including the exploitation of experimental results through sufficiently developed models, so as to obtain quantitative data for corrosion kinetics. In a very concrete way, the aim of our work was the development of an experimental procedure to investigate, using a laboratory pilot, the dynamics of corrosion on the surface of a steel plate, together with the evaluation of the results obtained through a phenomenological mathematical model. The model combines the equations of flow in the film with those of oxygen diffusion through the film and with the kinetics of the surface chemical reaction.

## 2. Materials and Methods

### 2.1. Experimental Setup and Procedures

When a film of water flows on a corroded surface, the flux of dissolved Fe, and therefore all the quantities that characterize this corrosion [10–13], can be obtained experimentally and supported theoretically. In principle, when corroding the film, if the water flow rate is high enough, then it is possible to have some $Fe^{2+}$ to $Fe^{3+}$ oxidation outside the surface with corrosion sites. Thus, if we recirculate the water that forms the film, then the increase in $Fe$ ions in the water will allow the expression of the removal dynamics of at least part of the ionized $Fe$ (dissolved, corroded, etc.) by the anodic reaction. The experimental facility was also built on this principle. We studied the corrosion dynamics over a longer period (several weeks) and created the assumed atmospheric environmental conditions similar to those of specific conditions over time. Water with an electrical conductivity of 100 µS/cm and an initial *pH* in the range of 6.7–7.1 was used as the initial corrosion medium and to fill the tank each time for the set of 5 long-term experiments for operating flow rates. The water used was selected to be not too different from the rainwater in terms of these two properties. The selected corrosion water flow rates were 1.9, 0.9, and 0.4 kg/min. Thus, the material exposed to the atmosphere was subjected to several hours of corrosion by the flow of the film, followed by its stopping when the corroding target plate dries under continuously monitored external humidity and temperature conditions, and then resuming the process. The experimental laboratory device, shown in Figure 3, was intended to show the flow of the film on the vertical steel plate where the corrosion occurs. The steel plate subjected to corrosion was 1000 cm in height, 150 cm in width, and 0.3 cm thick (Integrated Iron and Steel Works, Ukraine). Its composition is given in Table 1.

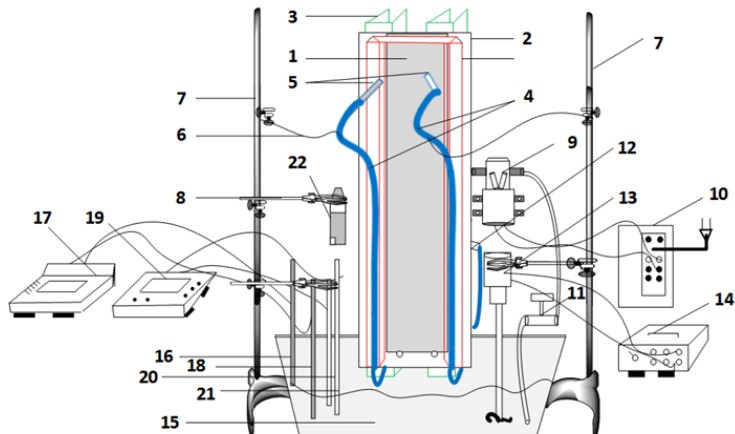

**Figure 3.** Experimental laboratory setup for corrosion through a water flowing film: (1) surface of black steel sheet; (2) plastic plate support; (3) PVC strengthening supports; (4) water feed pipes; (5) water nozzles; (6) water pipe flexible supports; (7) laboratory stands; (8) clamps stands; (9) water pump; (10) current transformer for currents up to 12 V; (11) water flow rate control valve; (12) splitter for water pipes; (13) laboratory liquid mixer; (14) electric power supply for liquid mixer; (15) water tank; (16) MV ion selective electrode; (17) Fischer ion/pH selective device; (18) conductivity electrode; (19) conductivity meter; (20) temperature probe; (21) pH electrode; (22) air humidity data logger.

**Table 1.** Composition of black steel sheet used in film corrosion research (according to manufacturer).

| Element | Composition (wt%) |
|---|---|
| Manganese (Mn) | 0.166 |
| Phosphorus (P) | 0.028 |
| Sulfur (S) | 0.028 |
| Carbon (C) | 0.206 |
| Chromium (Cr) | 0.078 |
| Molybdenum (Mo) | 0.114 |
| Vanadium (V) | 0.003 |
| Silicon (Si) | 0.004 |
| Copper (Cu) | 0.082 |
| Nickel (Ni) | 0.088 |
| Titanium (Ti) | 0.004 |
| Iron (Fe) | 0.199 |

Specifically, a corrosion kinetics experiment, with the presented installation, contains the following sequences:

1. Verification of the accuracy of the volume of liquid in the recirculation vessel of the corrosion medium (this must be exactly 5 L as initially determined);
2. Coordinating the start of all the equipment associated with the installation (conductivity meter, ion meter, humidity recorder, etc.);
3. Calibrating the workflow that forms the film on the plate;
4. Every 15 min, recording the dynamics of the electrical circuit conductivity of the corrosion medium, the state of corrosion of the plate (photo/successive images), and all external parameters (liquid temperature, average humidity data around the plate, etc.);
5. Taking a liquid sample to determine the total concentration of *Fe* in the corrosion medium;

6.   Stopping the film flow corrosion live simulation after 2–3 h to observe the drying dynamics of the plate and its condition for a new experiment.

Considering the measured values of *Fe* concentration in the corrosion medium (indirectly, by conductivity or directly, by chemical analysis using a spectrophotometric method [14]), the dynamics of the mean specific flow rate of dissolved iron ($N_{Fe}$) for an experiment is calculated. Equation (6) is thus used, where $c_{Ferz\ i}$ and $c_{Ferz\ i+1}$ are the iron concentrations of the corrosion medium at the corresponding times ($\tau_i$ and $\tau_{i+1}$), and $H$ and $l$ are the height and width of the liquid film on the corroding plate.

$$N_{Fe\ i} = \frac{V_l(c_{Fe\ rz\ i+1} - c_{Ferz\ i})}{H\ l(\tau_{i+1} - \tau_i)} \tag{6}$$

The corrosion dynamics measurements were performed at corrosion tank water flow rates of 1.9 kg/min, 0.9 kg/min, and 0.4 kg/min. The lower flow rate value was chosen so that the film formed on the plate remains stable, knowing that below the *Re* value of 140, the film can break [15,16]. From Table 2 we see that for this flow, the *Re* value was 174. The other two flow rates were chosen in such a way that their influence on the dynamics of the corrosion process can be highlighted as well as having an *Re* value below the limit of the turbulent flow in the film.

**Table 2.** Environmental conditions and data input for experimental investigation of steel corrosion in water falling film flow.

| Exp. No. | $Re_l$, $\delta$, $w_{max}$, $w$, $G_m$ | pH | $t_l$ °C | $t_a$ °C | $RH_a$ % | $\tau_d$ days | $\tau_e$ min | $c_{elrx0}\mu$ S/cm | $c_{Fe\ rz0}$ mg/L |
|---|---|---|---|---|---|---|---|---|---|
| 1 | 844.4 | 6.9 | 27.1 | 26.9 | 48.3 | - | 180 | 106 | 0 |
| 2 | $4.311 \times 10^{-4}$ m | 6.8 | 25.5 | 26.1 | 52.3 | 6 | 360 | 139 | 87 |
| 3 | 0.929 m/s | 6.8 | 26.5 | 26.4 | 33.8 | 6 | 360 | 151 | 140 |
| 4 | 0.619 m/s | 6.9 | 25.1 | 25.4 | 39.5 | 10 | 360 | 162 | 187 |
| 5 | 1.9 kg/min | 6.7 | 22.9 | 23.6 | 58.4 | 8 | 240 | 175 | 242 |
| 6 | 400 | 6.7 | 23.7 | 23.9 | 43.7 | 25 | 170 | 97 | 0 |
| 7 | $3.361 \times 10^{-4}$ m | 6.9 | 22.5 | 23.1 | 41.5 | 15 | 300 | 103 | 6 |
| 8 | 0.567 m/s | 6.9 | 21.3 | 21.7 | 53.2 | 9 | 360 | 120 | 38 |
| 9 | 0.368 m/s | 7.0 | 21.5 | 22.1 | 42.7 | 11 | 200 | 160 | 72 |
| 10 | 0.9 kg/min | 7.2 | 20.9 | 21.9 | 39.5 | 19 | 300 | 184 | 90 |
| 11 | 177.8 | 6.6 | 19.8 | 22.5 | 37.6 | 27 | 180 | 83 | 0 |
| 12 | $2.578 \times 10^{-4}$ m | 6.9 | 20.5 | 21.9 | 35.5 | 14 | 360 | 89 | 2.4 |
| 13 | 0.332 m/s | 6.9 | 20.8 | 21.6 | 55.1 | 14 | 300 | 95 | 5 |
| 14 | 0.221 m/s | 7.1 | 20.9 | 21.7 | 49.7 | 14 | 360 | 99 | 10 |
| 15 | 0.4 kg/min | 7.1 | 21.2 | 22.3 | 47.9 | 7 | 260 | 104 | 15 |

(pH) corrosion water; ($G_m$) mass flow; ($t_l$) corrosion water temperature; ($t_a$) air environment temperature; ($RH_a$) relative air humidity; ($\tau_d$) time from last experiment; ($\tau_e$) time of active experiment; ($c_{elrx0}$) initial corrosion water electrical conductivity; ($c_{Fe\ rz0}$) initial iron content of corrosion water in device tank.

## 2.2. Mathematical Modeling

Given the experimental basis, the modeling problem consists of finding a coherent set of relationships that express the dynamics of the concentration, $c_{Ferz}(\tau)$, and hence implicitly the dynamics of the corrosion evolution on the plate, on which the corrosion medium flows in the film ($N_{Fe}$ in Equation (6)). Figure 4 graphically introduces the description of the physical model that is considered here from the point of view of the transport, transfer, and conservation of oxygen species ($O_2$), without which, the evolution of corrosion cannot take place. Figure 4 describes a large part of the corrosion phenomenon on the plane plate. Thus, we see how the corrosion medium comes from the tank on the corrosion surface, with the oxygen concentration $c_{O2rz}(\tau)$, and flows like a piston with speed $w$ in a film of thickness $\delta$. This fact corresponds to the situation where the flow on the plane plate is considered to be simplified by the fact that it is of piston type with speed $w$, considered, since it is a laminar flow, to be 2/3 of the maximum surface velocity ($w_{max}$). Due to the evolution of

corrosion, at the current position $x$ along the corrosion surface where it consumes oxygen, the oxygen concentration drops to the average value ($c_{O2}$). For the control volume placed between $x$ and $x + dx$, there is an oxygen supply with a specific flow rate, $N_{O2}$; in fact, oxygen consumption due to the surface reaction evolves with the surface flow rate, $V_{rsO2}$.

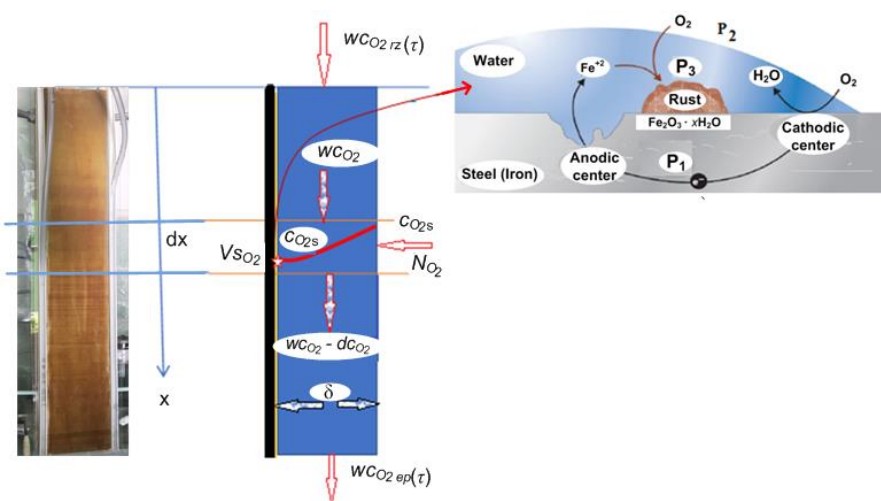

**Figure 4.** Vertical film flow corrosion model for black steel plate ($O_2$ balance, at the beginning of corrosion; $P_1$—electrons transfer between anodic and cathodic points, $P_2$—oxygen absorption, $P_3$—rust production).

Here, oxygen consumption determines:

1. The removal of iron from the metal surface, as a result of the surface reaction;
2. The formation of $Fe_2O_3 \cdot H_2O$ horst;
3. The release of $Fe$ ions into the corrosion medium.

Related to the control volume, the mentioned processes make the oxygen concentration at its exit $c_{O2}-dc_{O2}$. For the control volume, the oxygen balance for the unsteady state has the general form of Equation (7). This is transcribed into an analytical expression by Equation (8). Equation (9), which expresses the complete dynamics of the oxygen concentration near the surface subject to corrosion, results from dividing Equation (8) by the size of the control volume ($l\, \delta\, dx$).

$$\begin{pmatrix} Accumulated\ oxygen \\ flow\ rate\ into\ v.c \end{pmatrix} = \begin{pmatrix} Outgoing\ oxygen \\ flow\ rate\ from\ v.c \end{pmatrix} - \begin{pmatrix} Incomoing\ oxygen \\ flow\ rate\ in\ v.c \end{pmatrix} \quad (7)$$

$$l\delta dx\frac{\partial c_{O2}}{\partial \tau} = wl\delta(c_{O2} - dc_{O2}) - V_{rsO2}ldx - wl\delta c_{O2} + N_{O2}ldx \quad (8)$$

$$\frac{\partial c_{O2}}{\partial \tau} + w\frac{\partial c_{O2}}{\partial x} = \frac{1}{\delta}(N_{O2-}V_{rsO2}) \quad (9)$$

An addition to Equation (8) takes into account reverse flow, which occurs due to surface roughness or the formation of flow microwaves. It is phenomenologically characterized by the meaning of the axial mixing coefficient $D_L$. Completion of Equation (9) forms Equation (10). This form still needs to be completed, as shown below, by taking into account the consumption of oxygen by the oxidation of $Fe^{2+}$ ions.

$$\frac{\partial c_{O2}}{\partial \tau} + w\frac{\partial c_{O2}}{\partial x} = D_L\frac{\partial^2 c_{O2}}{\partial x^2} + \frac{1}{\delta}(N_{O2} - V_{rsO2}) \quad (10)$$

The model Equation (10) requires computational models or expressions for $\delta$, $w_{max}$, $w$, $D_L$, $N_{O2}$, and $V_{rsO2}$. We also need relationships or models that express the evolution of $Fe$ removal from the surface, the evolution of $Fe_2O_3$ horst growth, as well as the re-

lease of *Fe* ions into the corrosion medium (water flowing with the surface film). Some of these relationships are shown below. There are also data and comments related to other relationships.

When the mass flow rate is imposed by the film thickness (Equation (11)), the maximum superficial velocity (Equation (12)) and the mean velocity in the film (Equation (13)) are expressed by analogy with a laminar flow on a vertical plate [15,16] (where $\eta$ is the corrosion water viscosity; $g$ is the gravitational acceleration; and $\rho$ is the corrosion water density).

$$\delta = \sqrt[3]{\frac{3\eta G_m}{\rho^2 g l}} \tag{11}$$

$$w_{max} = \frac{\delta^2 g \rho}{2\eta} \tag{12}$$

$$w = \frac{2}{3} w_{max} = \frac{\delta^2 g \rho}{3\eta} \tag{13}$$

The specific oxygen flow rate is given by Equation (14), with the mass transfer coefficient expressed in Equations (15) and (16) as the mean value according to the penetration theory [17] or according to the Johnstone and Pigford model [18].

$$N_{O2} = k_l \left( c_{O2i} - c_{O2s} \right) \tag{14}$$

$$k_l = 2\sqrt{\frac{D_{O2} w_{max}}{\pi H}} \tag{15}$$

$$k_l = 3.41 \frac{D_{O2}}{\delta} \tag{16}$$

For the surface reaction rate, the Equations (1)–(4) are considered to occur in the active centers (Figures 2 and 4), which are located in the growing horst vicinity or/and under it. Experimental observations led to the conclusion that the appearance and development of corrosion centers follows an evolution both in time and along the *x-axis* direction. Then, for each active corrosion center a more or less complicated kinetics follows, for example, even first-order kinetics with respect to the local oxygen concentration. For a unit surface, the development of corrosion centers cannot exceed a maximum value, which is easily measured by photographing the surface and counting the number of horsts. In fact, the maximum fraction of surface area occupied by horsts ($\epsilon_{max}$) is obtained after a sufficient time from the start of corrosion in the film. Thus, the surface reaction rate, as kg $_{O2}/(\text{m}^2\text{s})$, is expressed by Equation (17), where $\alpha$ is a $T^{-1}$ dimensional constant, and the function $f(x,\tau)$ from Equation (18) seeks to describe the coverage of corrosion centers along the *x-axis*. It should also be noted that in the kinetic expression, the surface reaction rate constant ($k$) has the dimension $LT^{-1}$, that is, the dimension of a mass transfer coefficient. It is possible that the surface reaction rate constant also includes the resistance brought into the reaction due to the development of the rust film.

$$V_{sO2}(x,\tau) = \left(1 - e^{-\alpha\tau}\right) f(x) \epsilon_{max} k_{rs} c_{O2s} \tag{17}$$

$$f(x) = \begin{cases} 1 + \frac{x}{L} - \left(\frac{x}{L}\right)^2, & 1 - e^{-\alpha\tau} < 0.98 \\ 1, & 1 - e^{-\alpha\tau} \geq 0.98 \end{cases} \tag{18}$$

The flux of dissolved iron, in position $x$ and time $\tau$, becomes based on the stoichiometry of Equation (4). It is given by Equation (19). The stoichiometry in Equation (5) shows how we can establish the deposition flux of ferric hydroxide hydrate from Equation (20), which

is actually the flux showing rust development ($Fe_2O_3.zH_2O$). In Equation (20), the degree of $Fe^{2+}$ oxidation ($\eta_{rox}$) is a process parameter.

$$N_{Fe}(x,\tau) = \frac{4}{3}\frac{V_{sO2}(x,\tau)}{M_{O2}}M_{Fe} \tag{19}$$

$$N_{Fe(OH)3}(x,\tau) = \eta_{rox}N_{Fe}(x,\tau)\frac{M_{Fe(OH)3ZH20}}{M_{Fe}} \tag{20}$$

To determine how the iron ions reach the corrosion water (environment), it must be checked whether they ($pH$ = 6.9–7.2, $t$ = 25 °C) can react with the $Fe^{2+}$ ions as $Fe^{3+}$ ions. We thus start from the fact that $Fe(OH)_3$ has, at 291 K, a value of the solubility product $Ps$ = $1.1 \times 10^{-36}$. According to this $Ps$ value, it follows that from any location on the corrosion surface, where it is $Fe(OH)_3 \cdot zH_2O$ (actually $Fe_2O_3 \cdot zH_2O$), the concentration of trivalent iron ions on the corrosion surface is calculated according to the established Equations (21)–(24).

$$Fe(OH)_3 \leftrightarrow Fe^{3+} + 3OH^- \tag{21}$$

$$c_{OH} = 3c_{Fe3} \tag{22}$$

$$P_s = (c_{Fe3})(c_{OH})^3 = 27(c_{Fe3})^4 \tag{23}$$

$$c_{Fe3} = \left(\frac{P_s}{27}\right)^{0.25} = \left(\frac{1.1 \times 10^{-36}}{27}\right)^{0.25} = 4.5 \times 10^{-10}\frac{moli}{L} = 2.62 \times 10^{-8}\ g/L \tag{24}$$

This value of $c_{Fe3}$, namely $2.62 \times 10^{-8}$ g/L, makes the specific transfer flow to the corrosion medium around $2.62 \times 10^{-13}$ kg$_{Fe3}$/(m$^2$s). This low concentration of $Fe^{3+}$ in the corrosion medium cannot support the strong increase in its electrical conductivity. For a corrosion surface of 0.15 m$^2$ and 5 L of recirculated water, the flow rate of $Fe^{3+}$ ions coming out of the plate reaches $3.93 \times 10^{-14}$ kg$_{Fe3}$/s and the duration to obtain the $Fe^{3+}$ precipitation concentration in the corrosion medium is $\tau_p = (Vc_{Fe3})/(3.93 \times 10^{-14}) = 3.325 \times 10^3$ s. Our observations showed that this time of appearance of $Fe(OH)_3$ in the recirculated medium is shorter, not exceeding 2000 s. The explanation lies in the oxidation of $Fe^{2+}$ ions, released by corrosion, which reach this medium. According to Equation (20), we can write that the specific $Fe^{2+}$ flow coming into the corrosion medium can be expressed as Equation (25). Inside the corrosion environment, where oxygen exists, $Fe^{2+}$ is oxidized, as shown in Equation (26), to $Fe^{3+}$, which then precipitates as $Fe(OH)_3$.

$$N_{Fe2}(x,\tau) = (1 - \eta_{rox})N_{Fe}(x,\tau) \tag{25}$$

$$Fe^{2+} + \frac{1}{4}O_2 + \frac{1}{2}H_2O \rightarrow Fe^{3+} + OH^- \tag{26}$$

The kinetics of the homogeneous oxidation in water of $Fe^{2+}$ to $Fe^{3+}$ is expressed [19–21] in the form of Equation (27). Here, $PO_2$ is the partial pressure of oxygen in the air at the air/water interface and $c_{OH^-}$ and $c_{Fe2+}$ are the corresponding ion concentrations in the water. If we consider the partial pressure of $O_2$ air in the atmosphere, a temperature of 25 °C, a $pH$ interval of 6.5–7.5, and the ion concentrations expressed in mol/L, then the kinetic constant $k_{ox}$ has the value $3.33 \times 10^{11}$ L/(atm·mol·s).

$$-\frac{dc_{Fe2+}}{d\tau} = k_{ox}c_{Fe2+}(c_{OH^-})^2 P_{O_2} \tag{27}$$

If the corrosion medium also contains other important ions, such as $Na^+$ and $Cl^-$ (cases of corrosion in slightly saline water), or even different ones, then the elementary processes that accelerate corrosion must also be identified and, taking this into account, additions to Equations (19) and (28) must then be made accordingly. However, this is not the case in the current investigation. In our case, expressing $c_{OH^-}$ from the ionic product of water (Equation (22)), we will have, at 25 °C, atmospheric pressure, and pH = 7, a value for $c_{OH^-}$ of $10^{-7}$ mol/L. Now, taking $PO_2$ = 0.21 atmospheres, Equation (27) becomes Equation (28), a first-order kinetic relationship with respect to $c_{Fe2+}$, where the value $15.76 \times 10^{-3}$ s$^{-1}$ is for the reaction constant $k_{oxr}$. To express $c_{Fe2+}$ in kg$_{Fe2+}$/m$^3$, the right side of the expression is multiplied by the $Fe$ molecular mass.

$$-\frac{dc_{Fe^{2+}}}{d\tau} = v_{rFe^{2+}} = k_{oxr}c_{Fe^{2+}} \, M_{Fe} \tag{28}$$

The above data show that the $O_2$ balance must be completed, in relation to the control volume considered in Figure 4, with $O_2$ consumption for $Fe^{2+}$ oxidation. According to the stoichiometry of the reaction from Equation (26) and according to the $Fe^{2+}$ oxidation kinetics, the evolution of oxygen consumption in the corrosion medium has the expression from Equation (29). Figure 5 shows the mentioned completion and Equation (30) expresses the dynamics of the mean oxygen concentration in the film at position $x$ and time $\tau$. If it is admitted that the precipitation of $Fe(OH)_3$ is fast, then we will not have $Fe^{3+}$ in solution above $2.68 \times 10^{-8}$ g/L (see above). So, in the liquid phase, we have both $Fe^{2+}$ ions and $Fe(OH)_3$ micro particles resulting from $Fe^{2+}$ oxidation. Consequently, for the considered control volume, their balance must be added. Figure 6 shows the description of the $Fe^{2+}$ and $Fe(OH)_3$ balances.

$$v_{rO_2} = \frac{1}{4} \, v_{rFe^{2+}} \frac{M_{O2}}{M_{Fe}} \tag{29}$$

$$\frac{\partial c_{O2}}{\partial \tau} + w\frac{\partial c_{O2}}{\partial x} = D_L \frac{\partial^2 c_{O2}}{\partial x^2} + \frac{1}{\delta}\left(N_{O2} - V_{sO2} - N_{Fe(OH)3}\frac{M_{O2}}{4M_{Fe(OH)3ZH2O}}\right) - v_{rO_2} \tag{30}$$

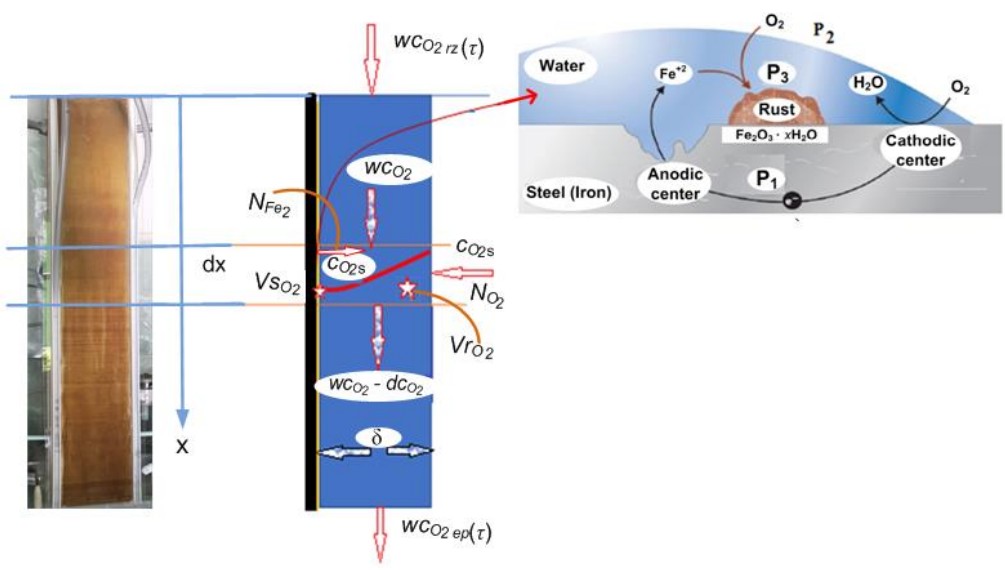

**Figure 5.** Vertical film flow corrosion model for steel plates (complete $O_2$ balance; $P_1$—electrons transfer between anodic and cathodic points, $P_2$—oxygen absorption, $P_3$—rust production).

The particularization of Equation (7) for $Fe^{2+}$ for the $Fe(OH)_3$ balance leads to the partial differential equations (Equations (31) and (32)), which express the dynamics of the

concentration of these species on the corroding plate. Here, we maintain the plug flow through axial mixing for the flowing liquid.

$$\frac{\partial c_{Fe^{2+}}}{\partial \tau} + w\frac{\partial c_{Fe^{2+}}}{\partial x} = D_L\frac{\partial^2 c_{Fe^{2+}}}{\partial x^2} + \frac{1}{\delta}N_{Fe2} - v_{rFe^{2+}} \tag{31}$$

$$\frac{\partial c_{Fe(OH)3}}{\partial \tau} + w\frac{\partial c_{Fe(OH)3}}{\partial x} = D_L\frac{\partial^2 c_{Fe(OH)3}}{\partial x^2} + \frac{M_{Fe(OH)3}}{M_{Fe}}v_{rFe^{2+}} \tag{32}$$

The general mathematical model of the process taking place on the plate contains the partial derivative equations (Equations (30)–(32)) with explanatory relationships for the reaction rate of oxygen consumption at the corrosion surface ($V_{sO2}$), for the specific oxygen flow rate to the corrosion surface ($N_{O2}$), for the reaction rate of oxygen consumption in the oxidation of $Fe^{2+}$ ($v_{rO2}$), for the specific flow rate of $Fe^{2+}$ coming into the liquid ($N_{Fe2}$), and for the oxidation rate of $Fe^{2+}$ in liquid ($v_{rFe2+}$).

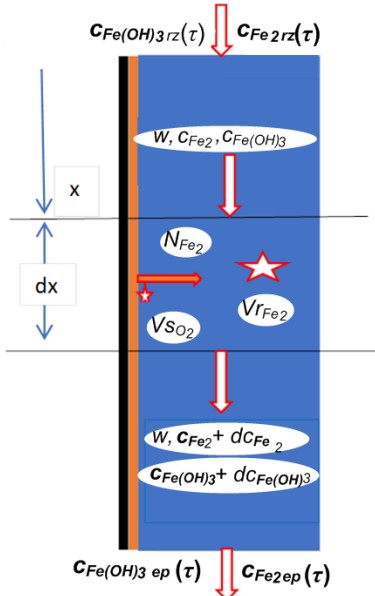

**Figure 6.** Vertical film flow corrosion model for steel plates ($Fe^{2+}$ and $Fe(OH)_3$ balance).

The model can be customized by attaching to it the univocity conditions expressed in Equations (33)–(35).

$$\tau = 0, \ x \geq 0, \ c_{O2} = c_{O2i}, \ c_{Fe^{2+}} = 0, \ c_{Fe(OH)3} = 0 \tag{33}$$

$$\tau > 0, \ x = 0, \ D_L\frac{dc_{O2}}{dx} = w(c_{O2\,rz}(\tau) - c_{O2\,rz\tau}), D_L\frac{dc_{Fe^{2+}}}{dx} = w(c_{Fe^{2+}}(\tau) - c_{Fe\,^{2+}rz\tau}), \ D_L\frac{dc_{Fe(OH)3}}{dx} = w\left(c_{Fe(OH)3}(\tau) - c_{Fe(OH)3\,rz\tau}\right) \tag{34}$$

$$\tau > 0, \ x = H, \ \frac{dc_{O2}}{dx} = 0, \ \frac{dc_{Fe^{2+}}}{dx} = 0, \ \frac{dc_{Fe(OH)3}}{dx} = 0 \tag{35}$$

It should be specified that Equation (34) is a univocity condition of the specified three-concentration field Equations (30)–(32) and a link relationship [19] between the corrosion model on the plane plate and the unsteady-state operation model of chemical processes in the corrosion medium tank. Regarding the processes that take place in the corrosion medium in the recirculation tank, Figure 7 tries to show their phenomenology. Here, we note that the perfect mixing flow model is accepted and that oxygenation of the corrosion medium occurs through the free surface. Also, the oxidation of $Fe^{2+}$ continues and due to

this oxidation, the generation of ferric hydroxide increases in intensity. This is measurable and easily observed by the increase in the yellow color of the corrosion medium.

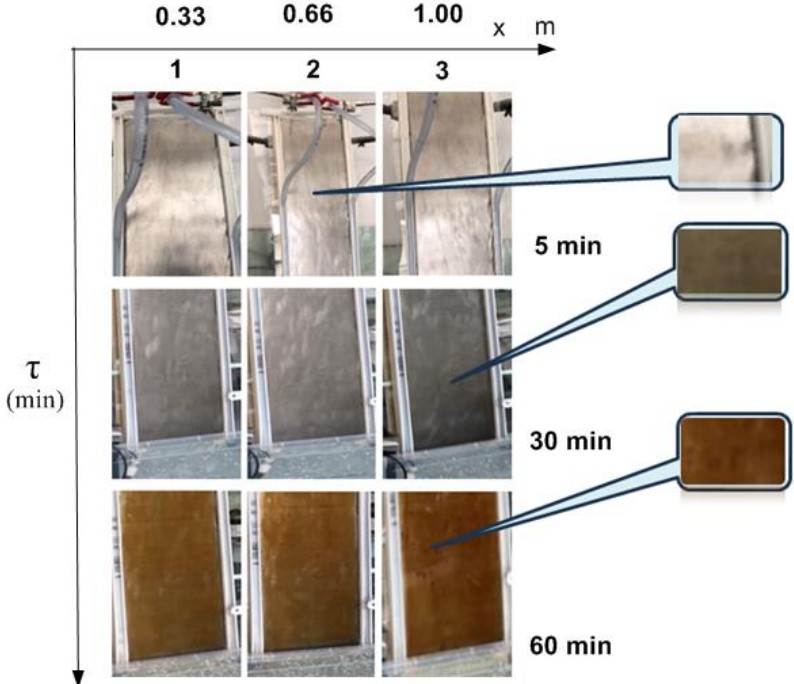

**Figure 7.** Corrosion evolution in time and space at the beginning of the process ($1 - x = (0 - 0.33$ m), $2 - x = (0.33 - 0.66$ m), $3 - x = (0.66 - 1$ m)).

The characteristic equations of the processes in the recirculation tank of the corrosion medium are the three unsteady-state balance equations for oxygen (Equation (36)), $Fe^{2+}$ ions (Equation (37)) and $Fe(OH)_3$ micro precipitate (Equation (38)). For these ordinary differential equations, the initial conditions in Equation (39) are used.

$$\frac{dc_{O2\,rz}}{d\tau} = \frac{G_{Vl}}{V_{rz}}\left(c_{O2\,ep} - c_{O2\,rz}\right) + \frac{S_{lb}}{V_{rz}}N_{O2\,rz} - \frac{1}{4}v_{rFe2rz}\frac{M_{O2}}{M_{Fe}} \tag{36}$$

$$\frac{dc_{Fe^{2+}rz}}{d\tau} = \frac{G_{Vl}}{V_{rz}}\left(c_{Fe\,ep^{2+}} - c_{Fe\,rz^{2+}}\right) + v_{rFe2rz} \tag{37}$$

$$\frac{dc_{Fe(OH)3\,rz}}{d\tau} = \frac{G_{Vl}}{V_{rz}}\left(c_{Fe(OH)3\,ep} - c_{Fe(OH)3\,rz}\right) + v_{rFe2rz}\frac{M_{Fe(OH)3}}{M_{Fe}} \tag{38}$$

$$\tau = 0,\ c_{O2rz} = c_{O2\,i} = 8\frac{mg}{L},\ c_{Fe^{2+}} = 0,\ c_{Fe(OH)3} = 0 \tag{39}$$

Therefore, the mathematical model of corrosion in film flow, coupled to the working mode of the experimental laboratory installation described above, contains the set of partial and ordinary derivative equations (Equations (30)–(39)). Added to this are the relationship expressions for $V_{sO2}$, $N_{O2}$, $v_{rO2}$, and $v_{rFe2+}$. The dynamics of the specific flow rate of dissolved iron ($N_{Fe}$), the dynamics of the specific flow rate of ferric hydroxide ($N_{Fe(OH)3}$), which is deposited as rust, and the dynamics of the specific flow rate of $Fe^{2+}$ ions ($N_{Fe2}$) are also of interest. They are related to each other and depend on the rate of the surface reaction ($V_{sO2}$).

There are 4 parameters in the model that require identification, namely $\alpha$, $k_{rs}$, and $\varepsilon_{max}$ in the $V_{sO2}$ expression (Equation (17)), and the $Fe^{2+}$ oxidation yield ($\eta_{ox}$) in the $N_{Fe(OH)3}$ expression (Equation (20)). It remains for the experimental investigation and for the numerical model software transcription to establish how these parameters will be identified

from the measured data describing the dynamics of the total *Fe* content and the electrical conductivity of the corrosion medium. Looking more generally at the above plate corrosion modeling during film flow with recirculating corrosion water, it is clear that the developed model is a phenomenological one based on transfer phenomena equations [22].

## 3. Results

We showed that the experimental investigation followed the film corrosion dynamics by measuring the electrical conductivity and the total iron content of the corrosion media for three flow rates (1.9 L/min, 0.9 L/min, and 0.5 L/min) on the plate. For each flow rate, five experiments were performed, with a duration between 3 and 6 h. Between experiments, the duration of the standby time was random even if the total flow duration was between 28 and 32 days. In the first experiment, with a water flow rate in the film of 1.9 L/min, special attention was paid to observing the beginning of corrosion as well as the state of the external factors of the process (environmental temperature, environment dew point, temperature of the corrosion environment, the humidity of the air in the vicinity of the plate). Figure 7 shows that the corrosion starts with an important evolution along the plate.

Figure 7 supports the considerations in Equation (14) regarding the coverage with corrosion centers of the surface on which the water film flows. We found that for $\tau = 1$ h, the surface of the iron plate was completely covered with active centers (Figure 7). Thus, $(1 - e^{-\alpha \tau})f(x)$ goes to 1, which imposes $\alpha = 3.91$ h$^{-1}$. This observation also completes and clarifies the initial and univocity conditions of the entire corrosion model. These are only valid for the first experiment, when it started with a new steel plate.

Following the coverage of the surface with active corrosion centers, the relation for $V_{sO2}(x,\tau)$ becomes as in Equation (40). Here the apparent surface reaction constant $k'_{rse}$ takes into it the fraction of surface occupied by horsts ($\varepsilon_{max}$) and the effect of blocking corrosion by increasing the thickness of the horst film surface.

$$V_{sO2}(x, \tau) = k'_{rse}c_{O2s}(x, \tau) \tag{40}$$

Figures 8 and 9 show that, in each of the 15 experiments associated with film flow rates of 1.9 L/min, 0.9 L/min, and 0.4 L/min, much care was taken to accurately measure the environmental factors at the time of the investigation. Table 2 summarizes the concrete conditions of each experiment at the beginning and during its evolutions.

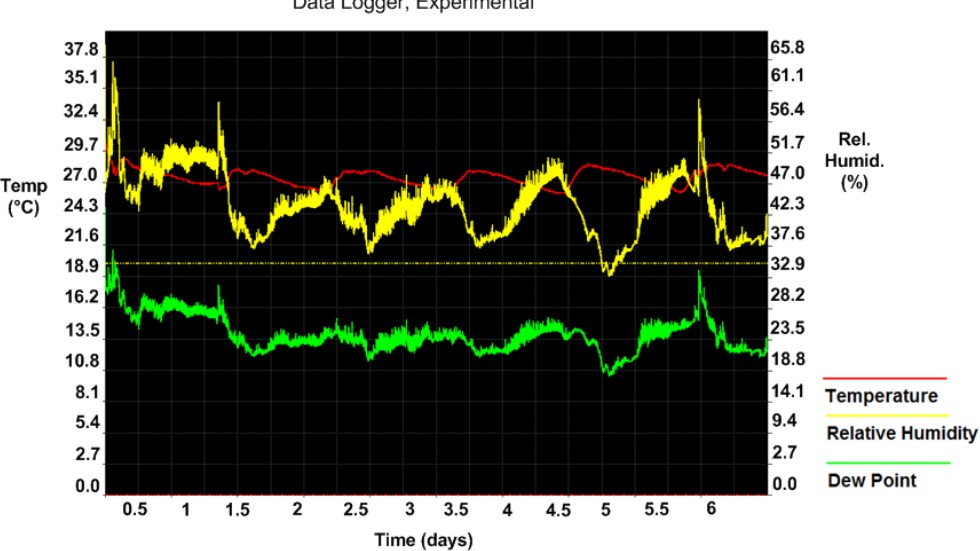

**Figure 8.** Dynamics of air parameters for the first corrosion experiment and for the time period until the second experiment (red—air temperature, yellow—relative air humidity, green—dew point temperature).

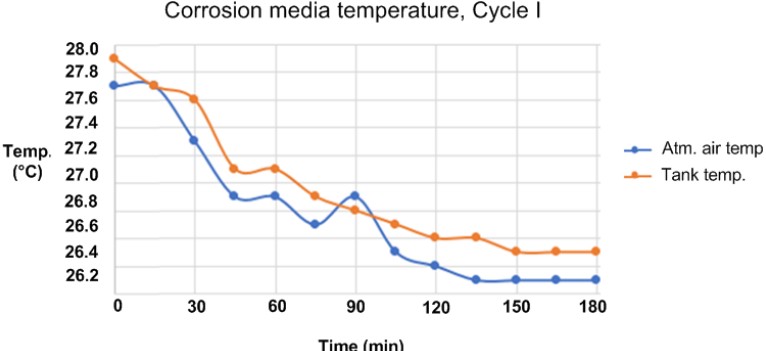

**Figure 9.** Dynamics of corrosion media temperature (red) and the surface air environment temperature (blue) for the first experiments at a corrosion flow rate of 1.9 L/min.

For each experiment, the dynamics of the electrical conductivity of the corrosion medium and the dynamics of its total *Fe* content are shown in Figures 10–15. First of all, for the purposes of this paper, we showed that each pair ($c_{el\ rz}$, $c_{Fe\ rz}$) of the 15 pairs of corrosion dynamics curves can be interpreted by the presented model, so that $k'_{rse}$ and $\eta_{rox}$ can be determined for each specific case. To justify the previous statement, we state that we have shown that the steel film corrosion model has four parameters, and we have shown that $\alpha$ has been identified and $\varepsilon_{max}$ has been merged into the surface reaction rate constant.

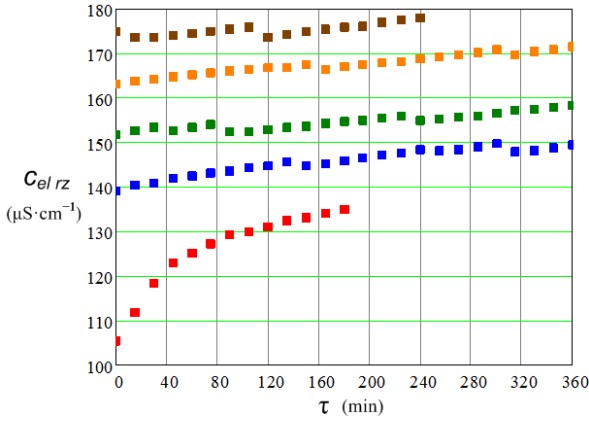

**Figure 10.** Dynamics of electrical conductivity of corrosion water from the 5 experiments with a liquid flow rate of 1.9 L/min ($Re_l$ = 844.4).

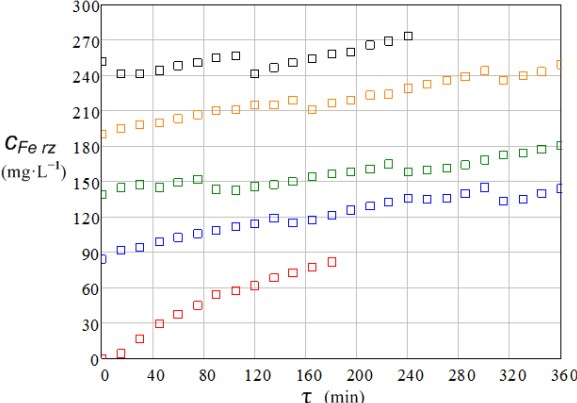

**Figure 11.** Dynamics of the total iron content of the corrosion water from the 5 experiments with a liquid flow rate of 1.9 L/min ($Re_l$ = 844.4).

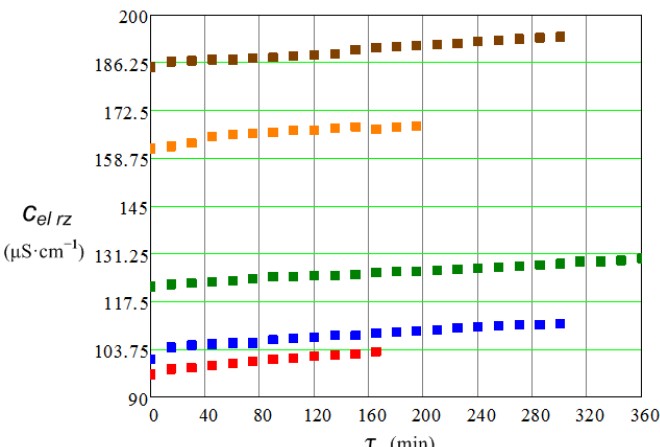

**Figure 12.** Dynamics of electrical conductivity of corrosion water from the 5 experiments with a liquid flow rate of 0.9 L/min ($Re_l$ = 400).

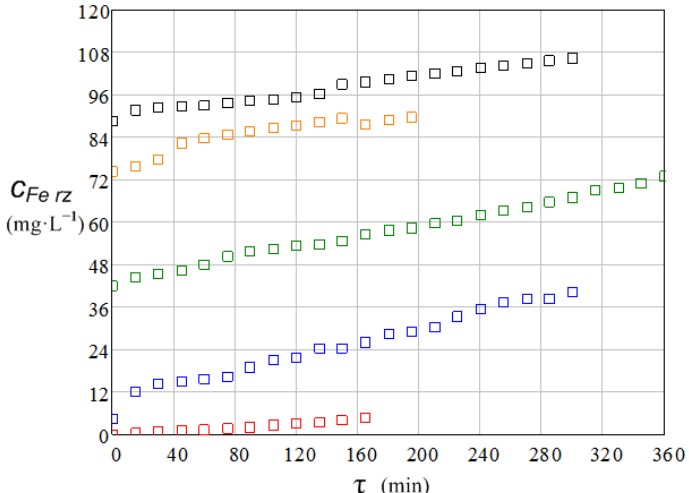

**Figure 13.** Dynamics of the total iron content of the corrosion water from the 5 experiments with a liquid flow rate of 0.9 L/min ($Re_l$ = 400).

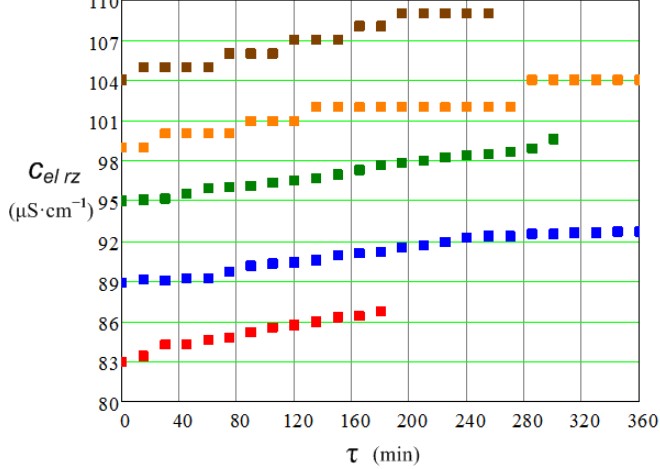

**Figure 14.** Dynamics of electrical conductivity of corrosion water from the 5 experiments with a liquid flow rate of 0.4 L/min ($Re_l$ = 177.8).

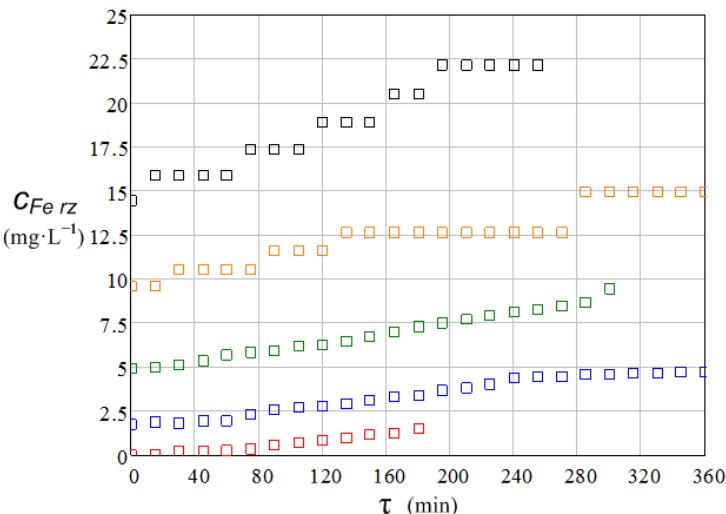

**Figure 15.** Dynamics of the total iron content of the corrosion water from the 5 experiments with a liquid flow rate of 0.4 L/min ($Re_l$ = 177.8).

Before turning to the problem of identifying the remaining basic parameters of the film flow steel corrosion model, it is noted that the data in Table 2 and Figures 11–15 show that:

- From the initial values of electrical conductivity, it follows that the experimental investigation took into account the case of corrosion in rainwater (Table 2 column $c_{elrx0}$);
- The place of investigation, a laboratory, was characterized by its own temperature conditions and relative air humidity (Table 2 columns $t_l$, $t_a$, and $RH_a$);
- The duration between experiments was random and, in total, covered almost half a year;
- The initial *pH* of the corrosion water and especially the fact that it did not change significantly during an active experiment shows that no processes occurred in addition to those considered in developing the model;
- There was a great similarity between the corresponding dependencies $c_{el\ rz}$ vs. $\tau$ and $c_{Fe\ rz}$ vs. $\tau$, which shows that the conductivity–concentration relationship is linear, as is often found at low ionic concentrations in solutions [23].

A quick look at the $c_{el\ rz}$ vs. $\tau$ and $c_{Fe\ rz}$ vs. $\tau$ curves can clearly distinguish that the first experiment that started with the virgin steel plate is characterized by the strongest dynamics (e.g., the $c_{el\ rz}$ increased by 45 μS/cm in 180 min and the $c_{Fe\ rz}$ has an increase of 82 mg/L in 180 min (red curve in Figures 10 and 11) versus an increase in $c_{el\ rz}$ of 11 μS/cm in 360 min and an increase in $c_{Fe\ rz}$ of 33 mg/L in 360 min (green curve in Figures 12 and 13)).

A comparison of the 15 experiments of active corrosion in the flowing film can be made by analyzing the dynamics of the flow of steel away from the plate, as well as its average maximum and minimum values. Thus, the data sets $c_{Fe\ rz}$ vs. $\tau$ were analytically transformed by a polynomial dependence of the 3rd degree and the specific corrosion flow rate, given by Equation (6), was expressed accordingly. Figure 16 and Table 3 show the result of this data processing. Table 3 also contains the data for the mean specific steel flow rate obtained with a linear relationship $c_{Fe\ rz}$ vs. $\tau$, which is supported by the shape of the curves shown in Figures 10–15.

Figure 16 and Table 3 show that:

1. The specific corrosion flow rate vs. the time dependences in Figure 16 decreased linearly and increased nonlinearly, with a maximum or a minimum due to the relationship of $c_{Fe\ rz}$ vs. $\tau$ by a polynomial with a 3rd degree of dependence;
2. It is interesting to note that the integral average values of the specific corrosion flow rate were very close to those where the dependence of $c_{Fe\ rz}$ vs. $\tau$ was linear (columns 5 and 6 of Table 3);

3.  The first experiment is distinguished, compared to all the others, by the dynamics of $c_{Fe\,rz}$ vs. $\tau$ and by the very high values of the specific flow; here, at $Re_l$ = 844.7 in 180 min, a $c_{Fe\,rz}$ of 80 mg/L was reached and the mean specific steel corrosion rate was 15.61 mg/(m²·min), compared to experiment 5 where $c_{Fe\,rz}$ increased in 240 min, with a $c_{Fe\,rz}$ of 30 mg/L with a mean flow rate specific to iron corrosion of 3.532 mg/(m²·min);

4.  These results support the previous observation which showed that the reduction of the specific flow of iron away from the plate is the consequence of the thickening of the rust layer on its surface.

None of the 15 reported experiments showed a sudden increase in the iron content in the corrosion medium, a fact that would show an important degradation of the rust film formed on the surface. Given the random distribution of time duration between experiments, this observation may be a sign that between two rainfall events, the rust layer on an attacked surface does not change its state or even its structure.

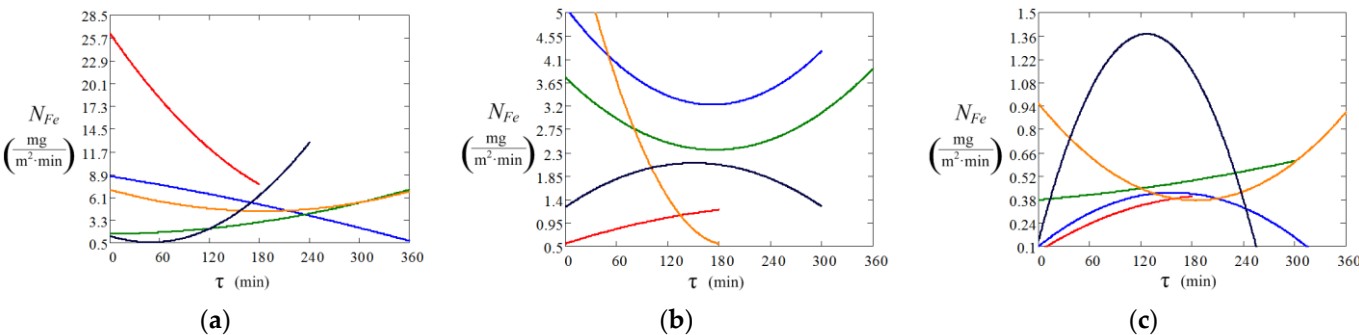

(a)          (b)          (c)

**Figure 16.** Dynamics of corrosion specific flow rate for experiments with (**a**) liquid flow rate of 1.9 L/min, (**b**) liquid flow rate of 0.9 L/min; (**c**) liquid flow rate of 0.4 L/min (colors correspond to those in Figures 10–15).

**Table 3.** Values for specific flow rate of iron for all active experiments.

| Exp. No. | Liquid Flow Rate | $N_{Fe\,min}$ mg/(m²·min) | $N_{Fe\,max}$ mg/(m²·min) | $N_{Fe\,mean}$[1] mg/(m²·min) | $N_{Fe\,mean}$[2] mg/(m²·min) |
|---|---|---|---|---|---|
| 1 | | 8.620 | 25.700 | 15.610 | 15.386 |
| 2 | | 0.550 | 8.970 | 5.060 | 5.115 |
| 3 | $G_{vl}$ = 1.9 L/min $Re_l$ = 844.4 | 1.900 | 7.220 | 3.470 | 3.325 |
| 4 | | 5.020 | 7.320 | 5.250 | 4.961 |
| 5 | | 0.550 | 14.250 | 3.880 | 3.353 |
| 6 | | 0.530 | 1.110 | 0.928 | 0.911 |
| 7 | | 3.150 | 5.150 | 3.728 | 3.567 |
| 8 | $G_{vl}$ = 0.9 L/min $Re_l$ = 400.0 | 2.390 | 3.890 | 2.849 | 2.676 |
| 9 | | 0.520 | 5.150 | 1.822 | 2.525 |
| 10 | | 1.370 | 2.150 | 1.822 | 1.918 |
| 11 | | 0.050 | 0.310 | 0.276 | 0.262 |
| 12 | | 0.110 | 0.380 | 0,323 | 0.323 |
| 13 | $G_{vl}$ = 0.4 L/min $Re_l$ = 177.8 | 0.380 | 0.550 | 0.478 | 0.475 |
| 14 | | 0.320 | 0.940 | 0.561 | 0.495 |
| 15 | | 0.170 | 1.360 | 0.996 | 1.091 |

[1] Integral mean value with 3rd degree polynomial dependence $c_{Fe\,rz}$ vs. $\tau$; [2] integral mean values with linear dependence $c_{Fe\,rz}$ vs. $\tau$.

If we were to report the specific corrosion flow rates measured in the case of steel industrial equipment, where water saturated with oxygen circulates (e.g., exchangers for industrial cooling), then we would have corrosion rates in mm/year in the range of 0.5–1.5 mm/year; therefore, it is rational to use deoxygenated water in such applications,

as well as an addition to the wall thickness of such equipment of 0.1 mm/year, which is used by designers.

Even though our data seem to show that reducing the water flow rate flowing on the steel surface leads to a reduction in the rate of the corrosion process, this fact must be viewed carefully because, here, the measurements of the flow rate reduction were made after the rust film had been established on the surface.

## 4. Discussion

Regarding the validation of the mathematical model through the experimental measurements presented above, the following sequence was used:

- The mathematical model was transposed into a numerical form as a function dependent on the parameters to be identified, namely $k'_{rse}$ and $\eta_{rox}$;
- The mentioned parameters were identified for all experiments using the minimization of the root mean square deviation (Equation (41)) between the values regarding the dynamics of the corrosion steel content in the medium that were calculated by the model and those measured experimentally [24,25];
- The value of $\eta_{rox}$ resulting from the experiment starting with the virgin steel plate was used as a baseline for all the other 14 measurements, so that they became faster and more accurate;
- From the identified values of the apparent surface reaction rate constant, the increase in oxygen mass transfer resistance due to the rust layer was determined by Equation (42), where $R_{sf\,r}$ is the surface reaction resistance and $R_{ru\,l}$ is the mass transfer resistance of the rust layer.

$$F\left(k'_{rse}, \eta_{ox}\right) = \sum_{i=1}^{N}\left(c_{Fe\,rz}\left(k'_{rse}, \eta_{ox}, \tau_i\right) - c_{Fe\,rz\,ex}\left(\tau_i\right)\right)^2 \tag{41}$$

$$R_T = R_{sf\,r} + R_{ru\,l} \tag{42}$$

Table 4 contains the values of the identified values for $k'_{rse}$ and $\eta_{ox}$ for all the developed experiments. In the numerical expression of $c_{Fe\,rz}$ ($k'_{rse}$, $\eta_{ox}$, and $\tau_i$), the axial dispersion coefficient was estimated to be $1.7 \times 10^{-6}$ m$^2$/s for $Re_l$ = 844.4, with a linear decrease for the other $Re_l$ values. It should also be noted that testing the sensitivity of the numerical model for the range $D_L$ $10^{-5}$–$10^{-7}$ m$^2$/s showed an extremely small influence of this parameter on the dynamics of the investigated process. The $k'_{rse}$ and $\eta_{ox}$ values, derived from the minimization of the mean squared deviation, have the quality of being maximum confidence values [25,26].

**Table 4.** The identified values of $k'_{rse}$, $\eta_{ox}$, and their external factors ($t_a$ and $t_l$: 21–27 °C, $RH_a$: 35–55%).

| Exp. No. | Liquid Flow Rate | Film Flow Time $\tau_f$ (h) | $\tau_d$ Days | $k'_{rse}$ $10^{-5}$ m/s | $\eta_{ox}$ - | $R_T$ $10^{-5}$ s/m | $R_{rul}$ $10^{-5}$ s/m | Figure 17 Case * |
|---|---|---|---|---|---|---|---|---|
| 1 | | 3 | - | 2.296 | 0.19 | 0.455 | 0 | (a) E1 |
| 2 | $G_{vl}$ = 1.9 L/min | 9 | 6 (6) | 0.716 | 0.18 | 1.397 | 0.941 | (a) E2 |
| 3 | $Re_l$ = 844.4 | 15 | 6 (12) | 0.506 | 0.16 | 1.976 | 1.521 | (a) E3 |
| 4 | | 21 | 10 (22) | 0.496 | 0.16 | 2.004 | 1.548 | (a) E4 |
| 5 | | 25 | 8 (30) | 0.358 | 0.16 | 2.793 | 2.338 | (a) E5 |
| 6 | | 28 | 25 (55) | 0.155 | 0.18 | 6.061 | 5.605 | (b) E1 |
| 7 | $G_{vl}$ = 0.9 L/min | 33 | 15 (70) | 0.425 | 0.18 | 2.353 | 1.897 | (b) E2 |
| 8 | $Re_l$ = 400.0 | 41 | 9 (79) | 0.336 | 0.17 | 2.976 | 2.521 | (b) E3 |
| 9 | | 44 | 11 (90) | 0.315 | 0.15 | 3.175 | 2.719 | (b) E4 |
| 10 | | 49 | 19 (109) | 0.298 | 0.17 | 3.356 | 2.899 | (b) E5 |
| 11 | | 52 | 27 (135) | 0.051 | 0.17 | 18.179 | 17.132 | (c) E1 |
| 12 | $G_{vl}$ = 0.4 L/min | 58 | 14 (149) | 0.032 | 0.14 | 29.411 | 28.861 | (c) E2 |
| 13 | $Re_l$ = 177.8 | 63 | 14 (163) | 0.043 | 0.16 | 23.258 | 22.796 | (c) E3 |
| 14 | | 69 | 14 (177) | 0.057 | 0.17 | 17.54 | 17.089 | (c) E4 |
| 15 | | 74 | 7 (184) | 0.106 | 0.16 | 9.434 | 8.978 | (c) E5 |

* From Figure 17: (a) E1–E5 experiments; (b) E1–E5 experiments; (c) E1–E5 experiments.

In Table 4, among the influencing factors on the two parameters, we also considered the liquid flow rate, the cumulative duration of film flow on the sample surface, as well as the rest duration (cumulative or non-cumulative) between corrosion dynamics experiments. The flow of the liquid influenced the supply of oxygen in the liquid film and on the surface of the rust film and implicitly at the corrosion centers. The cumulative flow duration is an indirect measure of whether the rust film on the corrosion surface is growing and thus controls the direction of oxygen in the kinetic process from the surface to corrosion. During the time between experiments, through drying and interaction with the adjacent air, changes in the structure of the rust film were possible, for example, in its hardening.

As expected, the apparent surface reaction constant showed high values when the rust film did not actually exist on the surface ($2.129 \times 10^{-5}$ m/s in experiment number 1). These values decreased close to 70 times when the rust film and its age were large ($0.032 \times 10^{-5}$ m/s in experiment number 12). The apparent constant for surface reaction correlates well (Table 5) with the influencing factors in Table 4. This fact quantitatively confirms that the process of surface film flow corrosion has a surface kinetic component, via the diffusion of oxygen through the rust film in the active process and the densification of the rust film (increasing the resistance of the structure to oxygen or to the diffusion of iron ions in the wet film) during the waiting period. Figure 17 shows the excellent coverage of the film corrosion dynamics measurements by the model using the parameter values in Table 4. Through these correlations, we have shown how the three factors influence the surface reaction rate constant and hence, the momentary corrosion speed. Many works have shown the influence of superficial flow [27,28] on the corrosion of steel or alloys. Similarly, in atmospheric corrosion, the rust layer formation on the corrosion surface reduces the corrosion speed ($v_c = kt^n$ with $n$ subunit [29]).

**Table 5.** Correlation coefficients for $k'_{rse}$ and $ln(k'_{rse})$ and process factor from Table 4.

| Correlation Coefficient | Reynolds Number $Re_l$ | Film Flow Time $\tau_f$ (h) | Standby Time $\tau_d$ (days) |
|---|---|---|---|
| $k'_{rse}$ | 0.649 | −0.712 | −0.649 |
| $lnk'_{rse}$ | 0.848 | −0.872 | −0.873 |

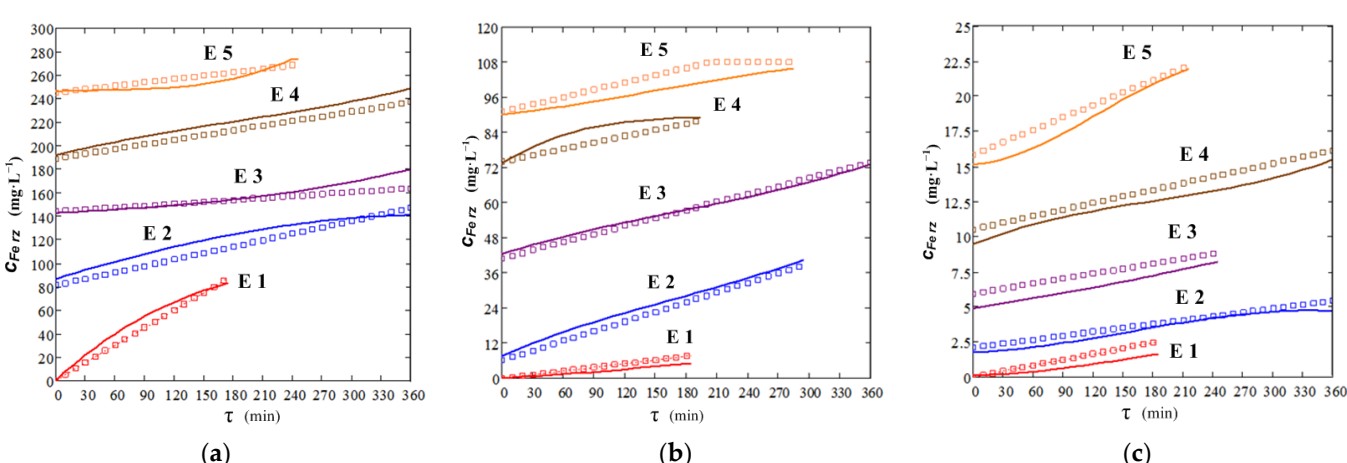

**Figure 17.** Dynamics of total iron content in corrosion water from experiments and according to the model, with parameters from Table 4 (*line*—experimental data, *points*—model computed data): (**a**) $Re_l = 844.4$, (**b**) $Re_l = 400$, (**c**) $Re_l = 177.8$, $t_a$ and $t_l$: 21–27 °C, $RH_a$: 35–55%.

The different behaviors of the rust film in the diffusion of oxygen to the surface reaction are highlighted by the graphical representation shown in Figure 18. Here, the mean thickness of the rust film in the experiment, $\delta_F$, and the effective oxygen diffusion coefficient came from the state of the numerical model when Equation (20) and the definition of $R_{ru\,l}$

are included. Two distinct ranges of values of the effective oxygen diffusion coefficient are identified in Figure 18. The range expressed by the mean values of the effective diffusion coefficient of $1.73 \times 10^{-9}$ m$^2$/s was characteristic for the time of the experiments in which the rust film did not adjusted its structure ($\tau_f$ below 50 h and $\tau_d$ above 120 days). The range of values with $D_{ef}$ around $0.29 \times 10^{-10}$ m$^2$/s corresponds to a much rigid (more compact) structure for oxygen diffusion to the reaction surface. The phenomenon of diffusion hardening of the rust film that we have identified can be supported by experimental analyses of the EDS and FTIR film that show, after tens and even hundreds of days, the change in its surface composition [30].

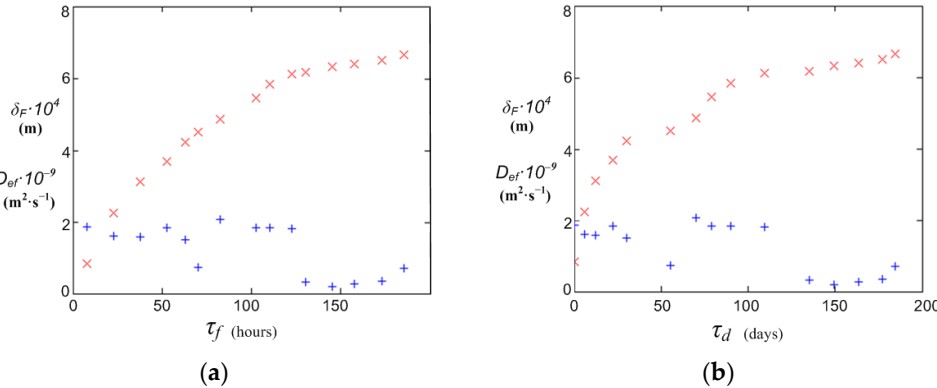

**Figure 18.** Rust film thickness and effective oxygen diffusion coefficient in terms of $\tau_f$ and $\tau_d$: (**a**) hours; (**b**) days.

The values identified for the oxidation yield of $Fe^{2+}$ on the steel plate were not influenced by the factors considered in Table 4. They can be characterized by the single value $\eta_{ox} = 0.167 \pm 0.012$. This result shows that iron oxidation occurs on the surface of the rust film where there is sufficient oxygen. Basically, the oxidation yield in film flow on a steel plate is, according to our measurements, in the range of 20–30 °C (Table 2).

Our investigation, which continued much longer than the times reported here, did not reveal surface rust removal by erosion due to the flow film.

## 5. Conclusions

A pilot laboratory device was developed along with the working procedure so that the surface corrosion dynamics of steel can be characterized when a corrosive medium is in the flow film.

The corrosion medium was water with electrical conductivity and *pH* close to rainwater, at flow rates chosen to be relevant and above the film flow stability limit.

A complex mathematical model with partial differential equations and ordinary differential equations for $O_2$, $Fe^{2+}$, and $Fe(OH)_3$ was considered to analyze the experimental data.

Numerical transposition of the model allowed the identification of model parameters, especially the apparent constant of surface corrosion rate and $Fe^{2+}$ oxidation yield surface in corrosion.

It was observed that the oxidation yield of iron ions on the surface of the rust film is not influenced by the intensity of the film flow on the surface and the age of the rust film.

Considering the dynamics of the oxygen diffusion coefficient identified by the rust film, it was found that, over time, the film formed on the surface becomes more and more resistant to oxygen diffusion, for which two distinct domains were identified.

**Author Contributions:** Conceptualization, M.C.I., T.D. and C.E.R.; methodology, M.C.I. and T.D.; validation, T.D., O.C.P., I.M. and I.M.D.; formal analysis, T.D., O.C.P., I.M. and I.M.D.; investigation, M.C.I. and T.D.; data curation, M.C.I., T.D. and C.E.R.; writing—original draft preparation, T.D. and M.C.I.; writing—review and editing, O.C.P. and C.E.R.; supervision, T.D. All authors have read and agreed to the published version of the manuscript.

**Funding:** This work has been funded by the European Social Fund from the Sectoral Operational Programme Human Capital 2014-2020, through the Financial Agreement with the title "Training of PhD students and postdoctoral researchers in order to acquire applied research skills—SMART", Contract no. 13530/16.06.2022—SMIS code: 153734.

**Institutional Review Board Statement:** Not applicable.

**Informed Consent Statement:** Not applicable.

**Data Availability Statement:** Not applicable.

**Conflicts of Interest:** The authors declare no conflict of interest.

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
