# Peer review of "Experimental Investigation and Modeling of Film Flow Corrosion"

_metals, doi:10.3390/met13081425_

Round 1

Reviewer 1 Report

The article is interesting and very useful, but needs to be revised.

1. Abstract
- The abstract needs to be rewritten. Currently, the first three sentences in the abstract are more general. Please make the abstract clearer. 

2. Introduction
- The introduction is well written, however, they should clearly state the purpose of the research at the end of the introduction.

3. Materials and methods
- In the experimental part, it is necessary to add the experiment environment. Even if distilled water was used as a corrosive environment, it should be written and the pH of the solutions should be indicated.
- From the results, it became clear that the authors used three flow velocities, while in the experimental part nothing is written about the selected values of flow velocities. It is necessary to write about it in the experimental methodology and explain why or give a reference to the selected flow rates.
- It is clear from Section 3 (Figure 7) that the authors performed corrosion tests, however, there is no information about these experiments in the methodological part. This information should be added to the methodology. 

4. Figures
- Please improve the quality for the following figures and make them more readable: 2,4-7,10-17.
- Figure 7: the quality needs to be improved for this figure. All figures included in Figure 7 should be brought to the same size. A scale bar should be indicated on the figures. In addition, it is clear from the text that the authors observed active corrosion centers... This does not follow from this figure. It looks very much like general corrosion is developing. I recommend to the authors to highlight localized metal dissolution centers in the figures and increase the size of the figures for clarity, if it is possible.  

5. Other comments
- From reading the results, it becomes clear that quite a large number of experiments were conducted. This does not follow from the methodological part. There is a lack of systematization of the data. I recommend to the authors to work on the systematization of information and its subsequent presentation. Now the material of the article is hard to perceive. The impression that the article is overloaded with data. 
- I recommend improving the conclusions. The first two conclusions should be rewritten. 
- I recommend to the authors to familiarize with the Template of this journal once again and according to these requirements to make Figures (Figure 9) and Tables (Tables 1-4, font size, captions of Tables).

Minor editing of English language required. 

Author Response

Thank you again for your comments and for you time.

A word file was uploaded. 

Best regards,

Authors

Reviewer 2 Report

The reviewed paper contains valuable information related to the detailed study of the mechanism of atmospheric corrosion. The presented measurement results have a practical aspect. Doubts concern only the general construction of the work. The kinetics of the described corrosion process depends mainly on the chemical composition of rainwater. This aspect has been practically omitted in the paper. Corrosion aggressiveness is largely related to the presence of CO2 in the atmosphere. The conditions for the formation of corrosion products partially having barrier properties for the above-mentioned reasons are very limited. The theoretical part of the work should include this important aspect together with the assessment of the impact of the location of the analysis of the occurrence of corrosion processes (marine, industrial, etc.). The pH of rainwater is usually in the range of 5.5-6.5, while in Table 2 this range is 6.8-7.2 ? The study did not provide an analysis of the composition of the water used in the research and possible sampling locations. The work is valuable, but needs to be completed in the current version.

Author Response

Thank you again for your comment and for your time.

A word file was uploaded.

Best regards,

Authors

Round 2

Reviewer 1 Report

Thank you to the authors for the work you have done! It can be seen that the authors tried very hard. However, there are still a number of comments that need to be addressed:

- Table 1: it is necessary to align the length of table rows. At the moment rows 2 and 3 are longer than the rest, please correct.

- Figures 2, 4, 5, 6, 8-11, 15, 17: quality and readability needs to be improved for this figure. The authors may have used an incorrect format for the figure. Please make appropriate changes.

- References: very many references that are outdated and need to be updated. Certainly, the problem is fundamental, so there should be references to earlier works. However, for the last 10-15 years clearly continued works in this direction. It would be good to update the list of references. Please make the appropriate changes.  

Nevertheless, the article is still interesting and relevant. I believe that after minor revision, this material can be published. 

Author Response

         Before moving on to the responses, we would like to thank you for your effort to go through our work with care and professionalism and for the observations and comments made on it so that it increases in quality and clarity.

Thank you again,

Best regards

Authors

Reviewer 2 Report

Corrections were made in the text of the thesis, I have no additional comments

Author Response

(The authors gave the same response as above.)
